# Automatic multi-anatomical skull structure segmentation of cone-beam computed tomography scans using 3D UNETR

**Maxime Gillot**[1,4]*, **Baptiste Baquero**[1,4], **Celia Le**[1,4], **Romain Deleat-Besson**[1,4], **Jonas Bianchi**[1], **Antonio Ruellas**[1], **Marcela Gurgel**[1], **Marilia Yatabe**[1], **Najla Al Turkestani**[1,10], **Kayvan Najarian**[1], **Reza Soroushmehr**[1], **Steve Pieper**[2], **Ron Kikinis**[3], **Beatriz Paniagua**[5], **Jonathan Gryak**[1], **Marcos Ioshida**[1], **Camila Massaro**[1], **Liliane Gomes**[1], **Heesoo Oh**[6], **Karine Evangelista**[1], **Cauby Maia Chaves Junior**[7], **Daniela Garib**[8], **Fábio Costa**[1], **Erika Benavides**[1], **Fabiana Soki**[1], **Jean-Christophe Fillion-Robin**[5], **Hina Joshi**[9], **Lucia Cevidanes**[1], **Juan Carlos Prieto**[9]

**1** University of Michigan, Ann Arbor, Michigan, United States of America, **2** ISOMICS, Cambridge, Massachusetts, United States of America, **3** Harvard Medical School, Boston, Massachusetts, United States of America, **4** CPE Lyon, Lyon, France, **5** Kitware, Clifton Park, New York, United States of America, **6** University of Pacific, Stockton, California, United States of America, **7** University of Ceara, Fortaleza, Brazil, **8** University of São Paulo, São Paulo, Brazil, **9** University of North Carolina at Chapel Hill, Chapel Hill, North Carolina, United States of America, **10** King Abdulaziz University, Jeddah, Saudi Arabia

* maxime.gillot@cpe.fr

**Data Availability Statement:** We have made all the code available at "https://github.com/Maxlo24/AMASSS_CBCT" that can be used to replicate this study. The database cannot be shared publicly

## Abstract

The segmentation of medical and dental images is a fundamental step in automated clinical decision support systems. It supports the entire clinical workflow from diagnosis, therapy planning, intervention, and follow-up. In this paper, we propose a novel tool to accurately process a full-face segmentation in about 5 minutes that would otherwise require an average of 7h of manual work by experienced clinicians. This work focuses on the integration of the state-of-the-art UNEt TRansformers (UNETR) of the Medical Open Network for Artificial Intelligence (MONAI) framework. We trained and tested our models using 618 de-identified Cone-Beam Computed Tomography (CBCT) volumetric images of the head acquired with several parameters from different centers for a generalized clinical application. Our results on a 5-fold cross-validation showed high accuracy and robustness with a Dice score up to 0.962±0.02. Our code is available on our public GitHub repository.

## 1 Introduction

Segmentation of medical and dental images is a visual task that aims to identify the voxels of organs or lesions from background grey-level scans. It represents a prerequisite for medical image analysis and supports entire clinical workflows from computer-aided diagnosis [1] to therapy planning [2], intervention [3], and follow-up [4]. Particularly for challenging dental and craniofacial conditions, such as dentofacial deformities, craniofacial anomalies, and tooth impaction, quantitative image analysis requires efficient solutions to solve the time-consuming

because the scans contain the patient facial skin that could allow facial recognition. Data are available from the University of Michigan Institutional Review Board (Contact Robert Eber, email reber@umich.edu) for researchers who meet the criteria for access to confidential data.

**Funding:** This work was supported by NIDCR R01 024450, American Association of Orthodontists Foundation Grabber Family Teaching and Research Award and by Research Enhancement Award Activity 141 from the University of the Pacific, Arthur A. Dugoni School of Dentistry. The funders had no role in the study design, data collection and analysis, decision to publish or preparation of the manuscript.

**Competing interests:** The authors have declared that no competing interests exist.

and user-dependent task of image segmentation. With medical and dental images being acquired at multiple scales and/or with multiple imaging modalities, automated image analysis techniques are needed to integrate patient data across scales of observation.

Due to the low signal/noise ratio of Cone-Beam CT (CBCT) images used in Dentistry, the current open-source tools for anatomic segmentation, such as ITK-SNAP [5] and 3D-Slicer [6] are challenging for clinicians and researchers. The large field of view CBCT images commonly used for Orthodontics and Oral Maxillofacial Surgery clinical applications require on average to perform detailed segmentation by experienced clinicians: (Fig 1) 7 hours of work for full face, 1.5h for the mandible, 2h for the maxilla, 2h for the cranial base (CB), 1h for the cervical vertebra (CV), and 30min for the skin. Additional challenges for accurate and robust automatic anatomical segmentation are the rich variety of anatomical structures morphology and the differences in imaging acquisition protocols and scanners from one center to another. Furthermore, patients that present with facial bone defects pose additional challenges for automatic segmentation because of unexpected anatomical abnormalities and variability. For this reason, the training of the machine learning models in the present study also included gold standard (ground-truth) clinicians' expert segmentations of CBCT images from patients with craniofacial large bone defects such as cleft lip and palate (CLP). Being able to accurately segment those maxillary deformities (Fig 1) is for the diagnosis and treatment planning of correction of the bone defects and craniomaxillofacial anomalies.

Although in the last decades, automatic approaches such as region seed growing [7], clustering methods, random forests [8], atlas-based system [9], and deep convolutional neural network (CNN) [10] have been proposed to segment the mandible, the maxilla, and the teeth, CBCT image segmentation remains challenging. Those previous studies focused on small samples from a single acquisition protocol; however, scans acquired at different clinical centers with different acquisition protocols, scales, and orientations require laborious manual

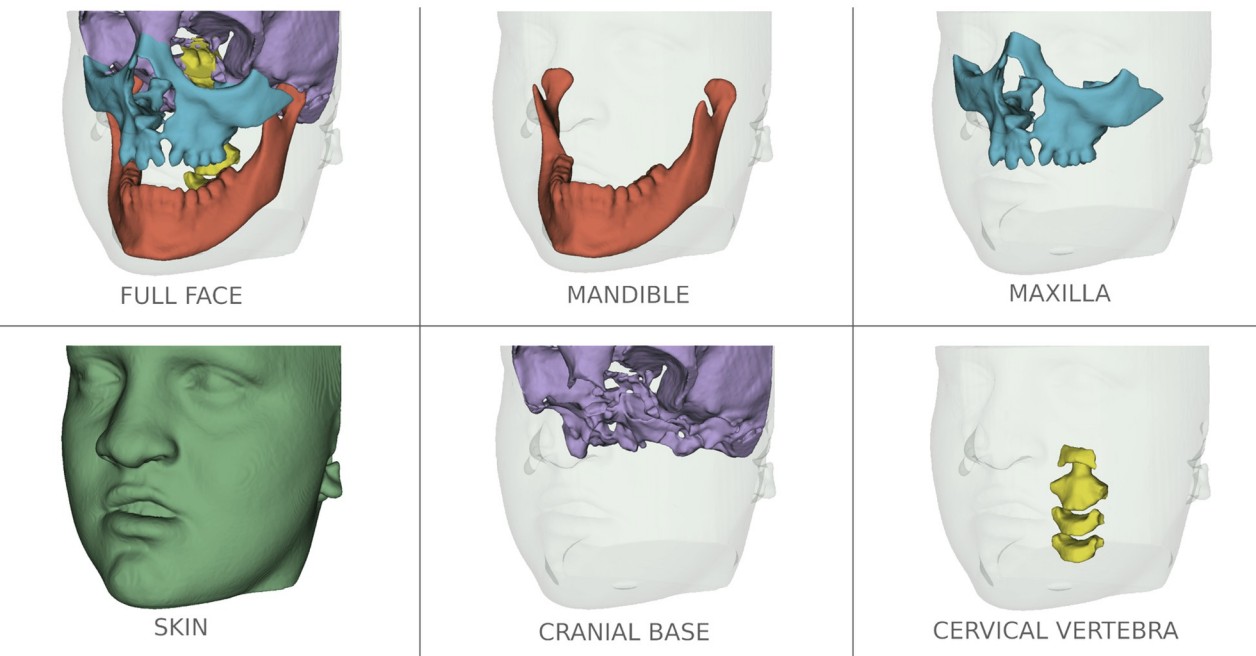

FULL FACE          MANDIBLE          MAXILLA

SKIN          CRANIAL BASE          CERVICAL VERTEBRA

**Fig 1. Multi-anatomical skull structure manual segmentation of the full-face by combining the mandible, the maxilla, the cranial base, the cervical vertebra, and the skin segmentation.** Patient has written consent on file for the use of the images.

correction in clinical settings to achieve accurate segmentation. Hence, methods for generalizable automatic image segmentation are sought.

The present study objective is to offer a free open-source tool to facilitate medical and dental image segmentation for clinics and research. We focused on the best practices for Artificial Intelligence in healthcare imaging across academia and enterprise researchers. Hence, the use of the new Medical Open Network for Artificial Intelligence (MONAI) framework that implements state-of-the-art machine learning algorithms such as the UNEt TRansformers (UNETR) [11]. In the following sections, we describe the data used to train our machine learning models, followed by related work on approaches to segment medical images, testing the performance of the proposed methods compared to the clinician's expert segmentation, and discussion of the novel results.

## 2 Materials

A total of 618 DICOM-formatted CBCT images of the head were used in this work. The images were acquired from 7 clinical centers with various scanners, image acquisition protocols, and field of views. All patient HIPAA identifiable personal information was removed from the DICOM files metadata through an anonymization process in the 3D Slicer platform [6]. The anonymization was performed before the clinical centers shared the data for this retrospective study. The University of Michigan Institutional Review Board HUM00217585 waived the requirement for informed consent and granted IRB exemption. Patients' skin was not removed from the large field of scans; however, those files are used only for the training of the proposed machine learning model.

Two open-source software packages, ITK-SNAP 3.8 [5] and 3D Slicer 4.11 were used by clinical experts to perform user interactive manual segmentation of the volumetric images and common spatial orientation of the head as the ground-truth to train our deep learning models.

All the 618 scans don't come with a full-face segmentation, the dataset was composed of 446 patients with mandible segmentation, 132 with the maxilla, 116 of the cranial base, 80 with the skin, and 14 patients with the cervical vertebra. The image spatial resolution varied from 0.16 to 0.5 mm$^3$ voxels. To test the robustness of the proposed method, patients with CLP were included in the dataset. Those patients have large bone defects in the jaw that varies a lot from one patient to another.

## 3 Related work

### 3.1 Region seed growing [7]

This method needs to place the seed inside the region of interest. The grayscale intensity grid and spatial distances from the seed to all the other voxels are computed to estimate a segmentation of similar features. This method showed less accuracy than the following methods and can require the clinicians to place the seeds.

### 3.2 Atlas-based system [9]

An atlas is defined as the combination of an intensity image and its segmentation to generate a template. From this point, 2 steps occur: label transfer which transfers segmentation labels from pre-labeled atlases to a novel image and label fusion which combines the label transfer results. The main con of this method is the lack of flexibility when exposed to high changes in the data such as in patients with CLP.

### 3.3 Random forests [8]

A probability grid is made to estimate the initial segmentation based on multiple expert-segmented CBCT images. The appearance features from CBCTs and the context features from the initial probability maps are both extracted to train a first-layer of random forest classifiers. A sequence of classifiers can segment CBCT images by iteratively training the subsequent random forest classifier using both the original CBCT features and the updated segmentation probability maps. Those methods are slow to train, computing-intensive and the prediction time can be high.

### 3.4 CNN

Previous methods where mostly using 2D [12] or 2.5D UNet [13], limited by computer power. Recent progress in GPU power and network architecture allowed the appearance of 3D CNN architectures showing better results than their 2/2.5D analogs. 3D UNet [10], TransUNet [14], and nnU-Net [15] showed high performance for medical imaging tasks including segmentation. However, the new UNETR architecture showed better results than all the previously cited CNN for CT segmentation.

## 4 Proposed method

Thanks to recent advances in deep learning, this study proposes a convolutional neural network (CNN) to extract a hierarchical feature representation for segmentation, which is robust to image degradation such as noise, blur, and contrast. Our algorithm requires Python 3.9 and uses various libraries to perform image processing. For the post-processing and the pre-processing, we are using ITK, SimpleITK, VTK, and connected-components-3d libraries. For the data augmentation and the segmentation, we used the MONAI library which simplifies the UNETR implementation and is optimized to process medical images in Python.

### 4.1 Pre-processing

Depending on the scanner and the image acquisition protocol, the CBCT scans are gray-scaled images with high contrast variation from one patient to another and the image spacing can be different. Among all the different spacing, 0.4 mm$^3$ is the most frequent. It's also a resolution that keeps enough details of the skull structure to segment while limiting memory usage with reasonable image size. From one center to another, the manual segmentation method can change. Different labels are used and the skull structure can be filled or not. From this point, to have more consistency in the dataset, all the data go through the following pre-processing steps:

- All the CBCTs and segmentations are re-sampled with a 0.4-mm$^3$ isometric voxel size using respectively a linear and a nearest-neighbor interpolation function.

- The scans go through a contrast adjustment function Fig 2. A cumulative graph is made from the image histogram ignoring the background color. The new minimum and maximum intensity are selected when reaching an $X_{min}$ and $X_{max}$ percentage on the cumulative graph. The intensity is then re-scaled in the [0, 1] interval.

- A "fill hole" morphological operation is applied to the segmentation and the label is set at 1.

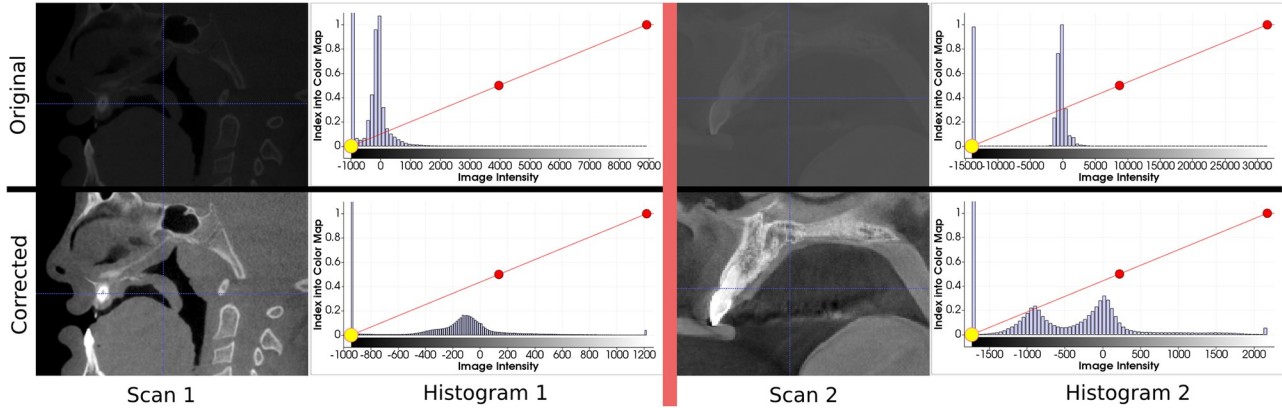

**Fig 2. Visualization of the contrast adjustment steps on two different scans.** This result is obtained by keeping the data between $X_{min}$ = 1% and $X_{max}$ = 99% on the cumulative graph.

## 4.2 UNETR

For this machine learning tool, we decided to use the new state-of-the-art model in 3D scan segmentation, the UNETR. Its architecture utilizes a transformer as the encoder to learn sequence representations of the input volume and effectively capture the global multi-scale information. The network design follows the successful "U-shape" for the encoder and decoder. The transformer encoder is directly connected to a decoder via skip connections at different resolutions to compute the final semantic segmentation output. The size of the scans to segment is not consistent and tends to be large (up to 600x600x600 voxels). No GPU is powerful enough to take this voxel grid size as input. We decided to shape our UNETR classifier with a 128x128x128 voxels input (Fig 3). To segment the entire image, the classifier moves across the scan to perform predictions in different locations. Once the entire image has been processed, segmented crops are merged to match the original input image size. Individual UNETR models were trained for different segmentation needs. All the models share common

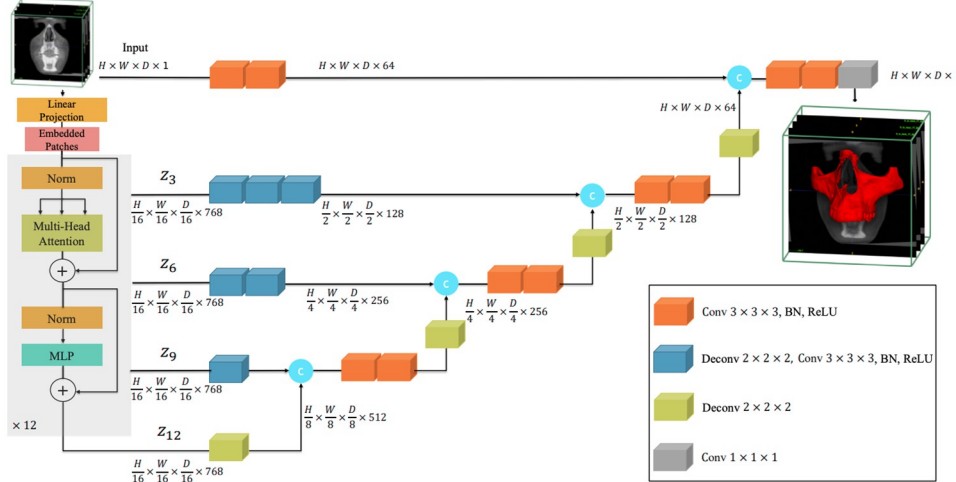

**Fig 3. Overview of the UNETR used.** A 128x128x128x1 cropped volume of the input CBCT is divided into a sequence of 16 patches and projected into an embedding space using a linear layer. A transformer model is fed with the sequence added with 768 position embedding. Via skip connections, the decoder will extract and merge the final 128x128x128x2 crop segmentation from the encoded representations of different layers in the transformer.

parameters: feature size = 16, hidden layer = 768, feedforward layer = 3072, number of attention heads = 12, and a dropout rate of 5%.

## 4.3 Training

For each skull structure to segment, the patients were sorted by separated folders based on the clinical center they were coming from. The dataset was then split into 3: 70% for the training, 10% for the validation, and 20% for testing. The data was split evenly from each folder to avoid overfitting to any specific center.

We used the MONAI "CacheDataset" tool to load the pre-processed data. Those datasets allow the use of transformers for data augmentations. Every time an image and its segmentation are loaded for the training, a number of $N_s$ cube samples are randomly cropped in the voxel grid. Those cubes all have the same $L_x \times L_y \times L_z$ shape to match the UNETR input size. For data augmentation (Table 1), random flip and 90° rotation are applied in each direction along with a random shift in intensity and contrast for the scans.

This step is applied to $N_i$ images to generate a batch of size $N_i \times N_s$. This batch is then fed into the UNETR the training. For the validation, data augmentation is also applied by only ignoring the cropping step, a prediction occurs on the full image using MONAI sliding window inference to move the UNETR classifier across the image. This network is optimized using the PyTorch library by a combination of a back-propagation algorithm to compute the network gradients and the Adam optimizer with weight decay. In this work, we used the weighted average of both the Dice loss (Table 1) and Cross Entropy Loss (Table 2) function.

$$DL = 1 - \frac{2\sum_{i=1}^{N} p_i g_i}{\sum_{i=1}^{N} p_i^2 + \sum_{i=1}^{N} g_i^2}, \tag{1}$$

where $p_i \in P$ is the predicted probability of the i-th voxel and $g_i \in G$ is the ground truth of the i-th voxel.

$$\ell(x, y) = L = \{l_1, \ldots, l_N\}^\top, \quad l_n = -w_{y_n} \log \frac{\exp(x_{n,y_n})}{\sum_{c=1}^{C} \exp(x_{n,c})}, \tag{2}$$

Where $x$ is the input, $y$ is the target, $w$ is the weight, $C$ is the number of classes, and $N$ spans the minibatch dimension as well as $d_1, \ldots, d_k$ for the $K$-dimensional case.

**Table 1. Data augmentation transformations for the training.**

| Data | Random crop | Random flip and rotation | Random shift in intensity | Random contrast adjustment |
|---|---|---|---|---|
| Images | Anywhere in the scan $N_s$ times | Along X, Y and Z-axis with a 25% probability for each axis for each axis | 50% chances of a 0.1 intensity shift | 80% chances to change image gamma in a [0.5,2] interval |
| Segmentation | | | N/A | N/A |

**Table 2. Comparison of manual and automatic segmentation using AUPRC, AUPRC-Baseline, Dice, F2 Score, Accuracy, Recall, and Precision of the 5-fold cross-validation for the 5 skull structures segmentation.**

| Structure | AUPRC | AUPRC Baseline | Dice | F2 Score | Accuracy | Recall | Precision |
|---|---|---|---|---|---|---|---|
| Mandible | 0.926 ± 0.037 | 0.011 ± 0.003 | 0.962 ± 0.020 | 0.961 ± 0.026 | 0.9992 ± 0.0005 | 0.960 ± 0.031 | 0.965 ± 0.026 |
| Maxilla | 0.738 ± 0.096 | 0.011 ± 0.003 | 0.853 ± 0.064 | 0.857 ± 0.061 | 0.996 ± 0.001 | 0.862 ± 0.073 | 0.855 ± 0.099 |
| Cranial base | 0.642 ± 0.127 | 0.018 ± 0.006 | 0.788 ± 0.103 | 0.804 ± 0.109 | 0.992 ± 0.004 | 0.824 ± 0.099 | 0.774 ± 0.135 |
| Cervical vertebra | 0.602 ± 0.145 | 0.008 ± 0.006 | 0.760 ± 0.113 | 0.723 ± 0.164 | 0.995 ± 0.004 | 0.704 ± 0.192 | 0.854 ± 0.033 |
| Skin | 0.947 ± 0.035 | 0.425 ± 0.72 | 0.971 ± 0.018 | 0.982 ± 0.009 | 0.974 ± 0.018 | 0.989 ± 0.009 | 0.954 ± 0.037 |

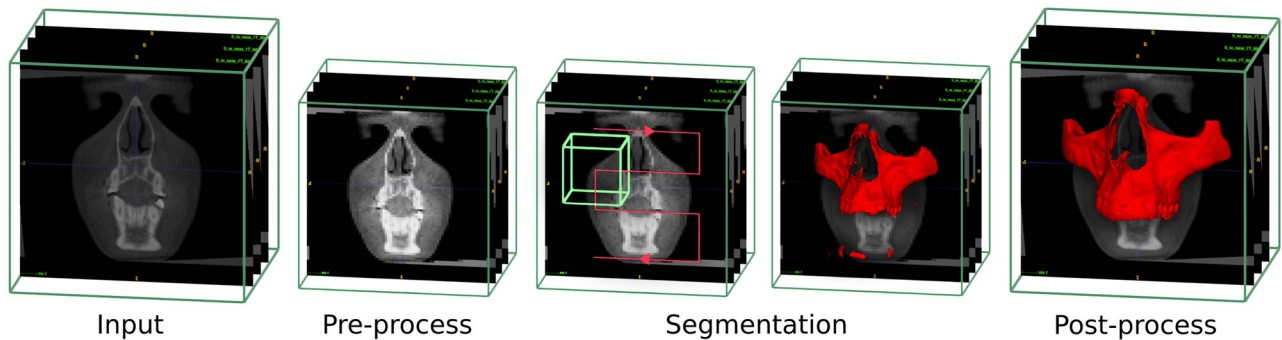

**Fig 4. Visualization of the automatic maxilla segmentation steps.** Re-sample and contrast adjustment of the input image, segmentation with the sliding window using UNETR, and finally, re-sampling of the cleaned-up segmentation to the input size.

The training was done on an NVIDIA Quadro RTX 6000/8000 GPU. With $X_{min} = 1\%$, $X_{max} = 99\%$, $L_x = L_y = L_z = 128$, $N_i = N_s = 10$ (batch size of 100), a dropout rate of 0.05, a learning rate of $10^{-4}$ and a weight decay of $10^{-5}$ it takes around 4h and 22GB of GPU memory for one model to be trained.

### 4.4 Segmentation and post-processing

Once we have a trained model, the challenge is to segment new scans that possibly have a different contrast and spacing than the ones used for the training. For the prediction, we create a new temporary file to work on and preserve the original. We apply the 2 first pre-processing steps (re-sample in a $0.4mm^3$ spacing if needed and adjust the contrast). The sliding window inference is then used to segment the whole image. We get as an output a voxel grid of probability on which we apply an argmax function. The segmentation can have some artifacts and unwanted elements. Therefore, we used the connected-components-3d 3.9.1 library [16] to keep the biggest segmented object only. A morphological operation is then applied to the segmentation to fill the holes. The final result is re-sampled to match the original image, orientation, spacing, origin, and size. All the steps are summarized in Fig 4.

## 5 Results

We performed a 5-fold cross-validation, each fold with a different 20% portion of the available data for the test. It allows testing the models on the entirety of the dataset.

The MONAI sliding window inference allows overlapping of the classifier for more precision but has a drastic impact on the computation time. During the validation step of the training, a prediction takes about 4s with 20% of overlap. To compute the metrics we used a 50% overlap to segment the test scans and it takes around 24s on GPU for each CBCT to be segmented. The prediction goes up to 1 min with an 80% overlap for even more precision.

To compare the clinician experts' manual segmentation and the AMASSS automatic segmentation, we used the Area Under the Precision-Recall Curve (AUPRC Eq 8) metric for class imbalance. Most of the bone groups represent about 10% of the volume only. Other metrics such as the recall (6), precision (7), Dice coefficient (DC Eq 3), and F2 (4) score were also computed to know how efficient the model is.

$$\mathrm{DC(M, A)} = \frac{2|A \cap M|}{|A| + |M|}, \tag{3}$$

Where M and A are respectively the binary image of the ground thruth segmentation and the

AMASSS output.

$$F_2 = \frac{TP}{TP + 0.2FP + 0.8FN}, \tag{4}$$

$$A = \frac{TP + TN}{TP + TN + FP + FN}, \tag{5}$$

$$R = \frac{TP}{TP + FN}, \tag{6}$$

$$P = \frac{TP}{TP + FP}, \tag{7}$$

Where TP stand for the number of true positive in the AMASSS output voxel grid, TN true negative, FP false positive and FN false negative.

$$\text{AUPCR} = \sum_{n=1}^{N-1} \frac{(R[n] - R[n-1]) \times (P[n] - P[n-1])}{2}, \tag{8}$$

Where R and P are the recall and the precision values from N confusion matrices for different thresholds. All these measurements (Table 2) vary from zero to one, where zero means no superposition between the two volumes, and one shows a perfect superposition between both. All metrics were performed on the binarized 3D images resulting from the post-processing. From a clinical point of view, it is better to have over-segmented images rather than under-segmented ones, and hence the F2 score was computed considering recall as twice as important as precision.

The average results for the mandible and the skin show the high precision of the automatic segmentations with a Dice above 0.96. Additionally, the standard deviation is quite low, indicating that the predictions are robust, consistent, and generalizable to unseen patients. Maxilla and cranial base showed similar results. The lower Dice compared to the mandible can be explained by fewer data used to train, but more importantly because of inconsistency from one ground-truth segmentation to another. The separation between the maxilla and the mandible can change, those regions have very thin bones and the amount of details segmented is different depending on the center. With only 14 segmentation available from one center, the cervical vertebra results are promising, showing the potential to be generalizable in future training with a larger sample.

We processed a full-face segmentation (Fig 5) of the patient Fig 1 that was kept out of all training. The CLP and even the cervical vertebra were successfully segmented, showing the robustness of the UNETR.

## 6 Discussion

This is the first study to our knowledge to use the new 3D UNETR architecture to segment multiple anatomic skeletal, dental, and soft tissue structures in the craniofacial complex of CBCT scans. Recent studies have focused on only one specific facial structure such as the maxilla, [17], mandible [18] or airway [12], and used smaller samples from a single CBCT acquisition protocol, thus, those algorithms are not yet generalizable like the proposed AMASSS.

Traditional image processing methods, such as super-voxels and graph clustering [19], atlas-based segmentation [8, 20], watershed methods [21] are available tools that presented good accuracy for segmentation, however, due to image artifacts and noise, that can be caused

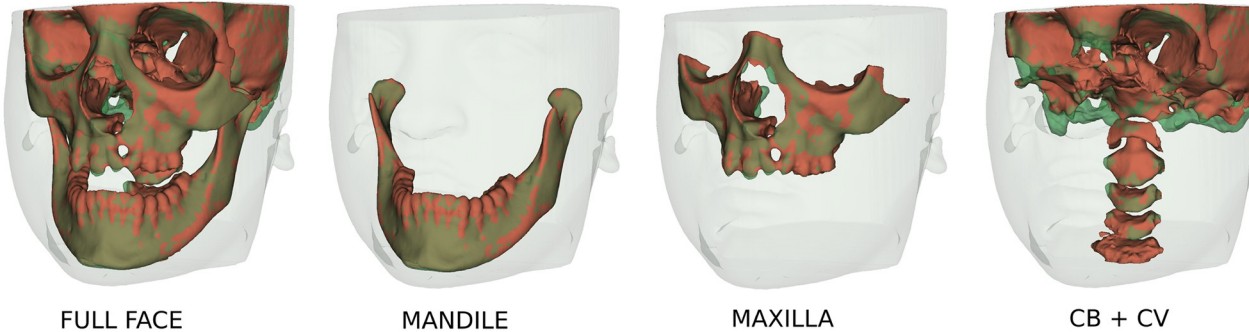

**Fig 5. Visualization of the automatic full-face segmentation results.** In red, the prediction is superposed with the manual segmentation in transparent green. On the full-face, we can see that the models managed to average the separation line between the maxilla and the mandible. The separation on the manual segmentation is different. It also explains why the metrics are lower than the mandible for those two skull structures.

by intercuspation of the dentition and the presence of metallic crowns, it is still a challenge to segment the images properly and also to segment different tissues such as bone with different densities (boundaries) and soft tissues. Due to these limitations, machine learning methods for image segmentation in dentistry have become popular, and the major limitation in training AI models such as the proposed AMASS is to have a gold standard to serve as training models [22]. To overcome this limitation, inthis study, manual annotations were performed for each scan used in the training, provided by clinicians with expertise and experience in 3D CBCT segmentations.

Moreover, AMASSS showed better and similar accuracy when compared to Si Chen et al.'s Maxilla segmentation with a dice score of 0.800 ± 0.029 and Verhelst et al.'s with a dice of 0.9722 ± 0.006 for the mandible segmentation, respectively. Commercial companies such as Materialise [23], Relu [24], and Diagnocat [25] have recently marketed AI-based segmentation for CBCT scans, but they are expensive and the precision of their algorithms require validation by clinicians.

Another important challenge in automated systems in dentistry, explained by Schwendicke et al. [26], is to provide solutions that can be largely entered into dental routine practice, and also follows principles such as demonstrating clinical value, protecting patient data, individual privacy, maintaining trustworthiness, and ensuring robustness and generalizability of the tools Towards these goals, the proposed open-source AMASSS algorithm was deployed as a free 3D Slicer extension "Automated dental tools". The software interface allows users to select the most updated trained model for increased precision of anatomic structures segmentation, continuously updating toward improved identification of patient facial structures and clinical applications [27].

Regarding the advantages and limitations, this study has the capacity of performing the segmentation of multi-structures in approximately 5 minutes; however, to achieve the necessary precision the ground-truth data can take several hours to be manually produced by the clinicians, which makes the addition of new structures of interest challenging and still human-dependent. Also, automated tools such as AMASS focus on future clinical decision support systems, to improve the human-computer interface rather than interrupt the clinical workflow [28], and for this reason, human interaction is still required, but less time-consuming.

Future work will continue to increase the databases for cervical vertebra, maxilla, and cranial base as well as add detailed anatomic structures such as the teeth roots and mandibular canals segmentation. Additional potential applications may be generalizable to other imaging

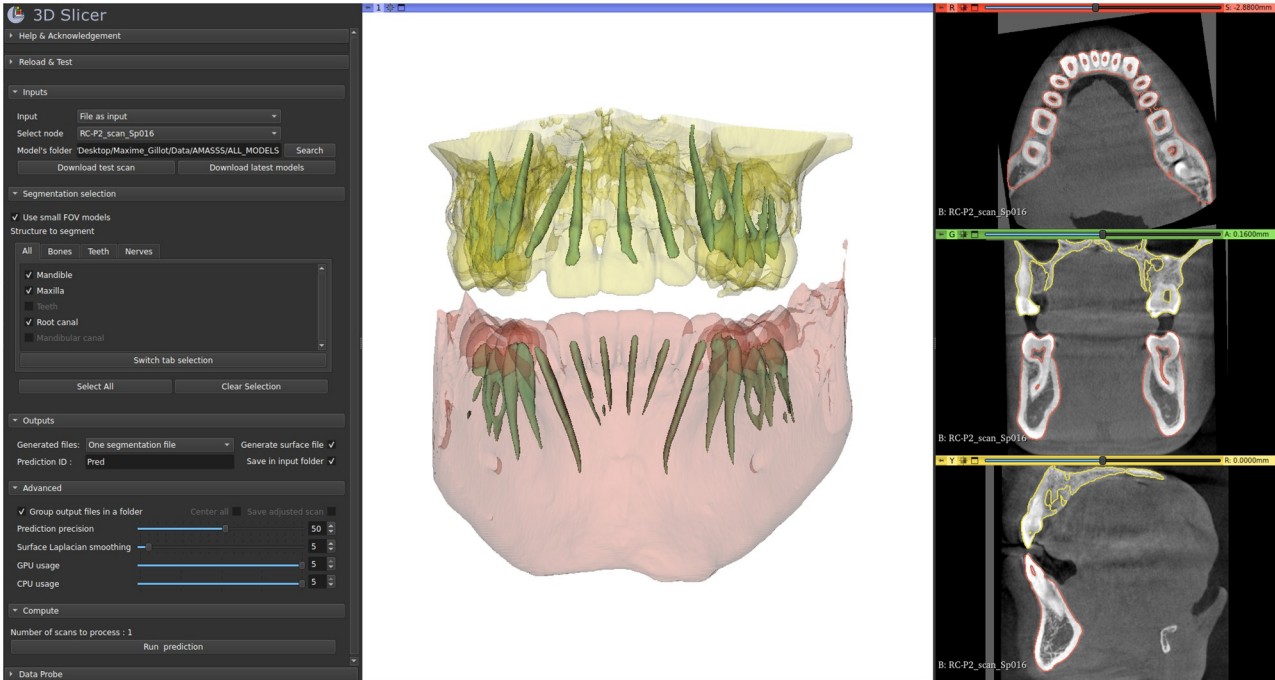

**Fig 6. 3D Slicer module in development for AMASSS-CBCT.** On the left, we can see the module with the different options/parameters. On the right, the visualisation of the segmentation applied on one small field of view scan with the selected skull structures. The mandible in red, the maxilla in yellow and the root canals in green.

modalities such as Magnetic Resonance Imaging, CT, micro CT, and ultrasound, which is been shown in recent manuscripts in the medical field [29, 30].

## 7 Conclusion

This proposal is a step towards the implementation of dentistry decision support systems, as machine learning techniques are becoming important to automatically and efficiently analyze dental images. The MONAI framework facilitated the processing of 618 CBCTs to perform fast training and data augmentation, which led to the high accuracy and robustness of the AMASSS tool. The UNETR showed high overall performance, achieving a Dice up to 0.962 ± 0.02 on heterogeneous CBCT images.

Given its robustness and performance time, this validated free tool was implemented in 2 open-source ecosystems, a web-based clinical decision support system (the Data Storage for Computation and Integration, DSCI) [31], and a user-friendly 3D Slicer module Fig 6. These computer-aided diagnostic tools will aid in diagnosis and therapy planning, especially for patients with craniomaxillofacial anomalies and deformities.

## Author Contributions

**Conceptualization:** Jonas Bianchi, Antonio Ruellas, Lucia Cevidanes.

**Data curation:** Maxime Gillot, Celia Le, Romain Deleat-Besson, Jonas Bianchi, Marcela Gurgel, Marilia Yatabe, Najla Al Turkestani, Jonathan Gryak, Marcos Ioshida, Camila Massaro, Liliane Gomes, Heesoo Oh, Karine Evangelista, Cauby Maia Chaves Junior, Daniela Garib, Fábio Costa, Erika Benavides, Fabiana Soki.

**Formal analysis:** Maxime Gillot, Antonio Ruellas, Marcela Gurgel, Marilia Yatabe, Najla Al Turkestani, Jonathan Gryak, Marcos Ioshida, Camila Massaro, Liliane Gomes, Heesoo Oh, Karine Evangelista, Cauby Maia Chaves Junior, Daniela Garib, Fábio Costa, Erika Benavides, Fabiana Soki, Lucia Cevidanes.

**Funding acquisition:** Lucia Cevidanes.

**Investigation:** Maxime Gillot, Antonio Ruellas, Marcela Gurgel, Marilia Yatabe, Najla Al Turkestani, Jonathan Gryak, Marcos Ioshida, Camila Massaro, Liliane Gomes, Heesoo Oh, Karine Evangelista, Cauby Maia Chaves Junior, Daniela Garib, Fábio Costa, Erika Benavides, Fabiana Soki, Lucia Cevidanes.

**Methodology:** Maxime Gillot, Antonio Ruellas, Lucia Cevidanes, Juan Carlos Prieto.

**Project administration:** Maxime Gillot, Antonio Ruellas, Lucia Cevidanes.

**Software:** Maxime Gillot, Baptiste Baquero, Steve Pieper, Ron Kikinis, Hina Joshi, Juan Carlos Prieto.

**Visualization:** Maxime Gillot.

**Writing – original draft:** Maxime Gillot.

**Writing – review & editing:** Baptiste Baquero, Celia Le, Romain Deleat-Besson, Jonas Bianchi, Antonio Ruellas, Marcela Gurgel, Marilia Yatabe, Najla Al Turkestani, Kayvan Najarian, Reza Soroushmehr, Steve Pieper, Ron Kikinis, Beatriz Paniagua, Jonathan Gryak, Marcos Ioshida, Camila Massaro, Liliane Gomes, Heesoo Oh, Karine Evangelista, Cauby Maia Chaves Junior, Daniela Garib, Fábio Costa, Erika Benavides, Fabiana Soki, Jean-Christophe Fillion-Robin, Hina Joshi, Lucia Cevidanes, Juan Carlos Prieto.

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
