## [Decision Letter · Decision Letter 0]

24 Jun 2022

PONE-D-22-13724Automatic multi-anatomical skull structure segmentation of cone-beam computed tomography scans using 3D UNETRPLOS ONE

Dear Dr. Gillot,

Thank you for submitting your manuscript to PLOS ONE. After careful consideration, we feel that it has merit but does not fully meet PLOS ONE’s publication criteria as it currently stands. Therefore, we invite you to submit a revised version of the manuscript that addresses the points raised during the review process.

We look forward to receiving your revised manuscript.

Kind regards,

Sathishkumar V E

Academic Editor

PLOS ONE

Journal Requirements:

"Supported by NIDCR R01 024450, AA0F Dewel Memorial Biomedical Research award and by Research Enhancement Award Activity 141 from the University of the Pacific,Arthur A. Dugoni School of Dentistry."

"The authors received no specific funding for this work."

Reviewers' comments:

Reviewer's Responses to Questions

**Comments to the Author**

1. Is the manuscript technically sound, and do the data support the conclusions?

Reviewer #1: Yes

Reviewer #2: Partly

2. Has the statistical analysis been performed appropriately and rigorously? 

Reviewer #1: Yes

Reviewer #2: No

3. Have the authors made all data underlying the findings in their manuscript fully available?

Reviewer #1: Yes

Reviewer #2: No

4. Is the manuscript presented in an intelligible fashion and written in standard English?

Reviewer #1: Yes

Reviewer #2: No

5. Review Comments to the Author

Reviewer #1: I have read the manuscript “Automatic multi-anatomical skull structure segmentation of

cone-beam computed tomography scans using 3D UNETR”. I have following comments.

1. The literature suvery is poor and the relevent algorithms are required to be cited in the paper.

2. The introduction of the proposed coarse-to-fine framework is simple, especially about the patch-based semantic segmentation. As the main method of this paper, it would be better to give a more detail explanation about the coarse-to-fine framework and patch-based semantic segmentation.

3. The results from the relevent algorithms (of airway segmentation) are required to be compared with the prposed method results.

4. U-Net Architecture was used in the manuscript. However, authors didn't disclose the details of the deep learning architecture.

5. No result was shown for the comparison of manual and authomatic segmentation which was claimed through the title. The results are required which can show the error/difference between the manual volume assessment and automatic volume assessment.

6. Again Literature is incomplete. There are many more articles on CBCT based volumetric segmentation and written much elaborately. Should have had a proper comparative table on literature and flow chart giving the flow of data analysis.

7. Results are not discussed with any available literature. There lacks of conclusions and analysis of the experimental results in the results section. It would be more convincible if the authors can give additional technical analysis about their experimental results.

Reviewer #2: 1.Introduction section needs to be re-written to improve its quality and readability.

2.What is the motivation of the proposed work? Research gaps, objectives of the proposed work should be clearly justified

3.Overall, the basic background is not introduced well, where the notations are not illustrated much clear.

4.The literature has to be strongly updated with some relevant and recent papers focused on the fields dealt with in the manuscript.

5.The study lacks a theoretical framework which is important for the reader to grasp the crust of the research.

6. Explain why the current method was selected for the study, its importance and compare with traditional methods.

7.Authors are suggested to include more discussion on the results and also include some explanation regarding the justification to support why the proposed method is better in comparison towards other methods

8.Does this kind of study have never attempted before? Justify this statement and give an appropriate explanation to do so in this paper.

9.Quality of figures is so important too. Please provide some high-resolution figures. Some figures have a poor resolution.

10.The language usage throughout this paper need to be improved, the author should do some proofreading on it.

6. PLOS authors have the option to publish the peer review history of their article (what does this mean?). If published, this will include your full peer review and any attached files.

Reviewer #1: No

Reviewer #2: No

---

## [Author Response · Author response to Decision Letter 0]

7 Sep 2022

Response to reviews

Thank you for your careful consideration of our study for publication in the PLOS ONE Journal. We have addressed each of the reviewer’s constructive comments and Journal requirements that have strengthened this submission

Journal Requirements:

https://journals.plos.org/plosone/s/file?id=ba62/PLOSOne_formatting_sample_title_authors_affiliations.pdfAnswer: We have ensured that the manuscript and file naming meets PLOS ONE's style requirements following your sdtyle templates. 

2. Please provide additional details regarding participant consent. In the ethics statement in the Methods and online submission information, please ensure that you have specified (1) whether consent was informed and (2) what type you obtained (for instance, written or verbal, and if verbal, how it was documented and witnessed). If your study included minors, state whether you obtained consent from parents or guardians. If the need for consent was waived by the ethics committee, please include this information.If you are reporting a retrospective study of medical records or archived samples, please ensure that you have discussed whether all data were fully anonymized before you accessed them and/or whether the IRB or ethics committee waived the requirement for informed consent. If patients provided informed written consent to have data from their medical records used in research, please include this information.

Answer: We have now clarified in the methods section that: ”All patient HIPAA identifiable personal information was removed from the DICOM files metadata through an anonymization process in the 3D Slicer platform . The anonymization was performed before the clinical centers shared the data for this retrospective study.The University of Michigan Institutional Review Board HUM00217585 waived the requirement for informed consent and granted IRB exemption. The patients' skin of the large field of scans was not removed; however those files are used only for training of the proposed machine learning model.”

For figure 1, the patient has written consent on file for the use of the images acquired for clinical purposes.

3 and 4. We note that the grant information you provided in the ‘Funding Information’ and ‘Financial Disclosure’ sections do not match. When you resubmit, please ensure that you provide the correct grant numbers for the awards you received for your study in the ‘Funding Information’ section.

Thank you for stating the following in the Acknowledgments Section of your manuscript: 

"Supported by NIDCR R01 024450, AA0F Dewel Memorial Biomedical Research award and by Research Enhancement Award Activity 141 from the University of the Pacific,Arthur A. Dugoni School of Dentistry."

"The authors received no specific funding for this work." Please include your amended statements within your cover letter; we will change the online submission form on your behalf.

Answer: Thank you for your guidance. We have now removed the funding-related text from the manuscript and added to the cover letter the following statement:” This work was supported by NIDCR R01 024450, American Association of Orthodontists Foundation Dewel Memorial Biomedical Research award and by Research Enhancement Award Activity 141 from the University of the Pacific, Arthur A. Dugoni School of Dentistry.”

5. In your Data Availability statement, you have not specified where the minimal data set underlying the results described in your manuscript can be found. PLOS defines a study's minimal data set as the underlying data used to reach the conclusions drawn in the manuscript and any additional data required to replicate the reported study findings in their entirety. All PLOS journals require that the minimal data set be made fully available. For more information about our data policy, please see http://journals.plos.org/plosone/s/data-availability.Upon re-submitting your revised manuscript, please upload your study’s minimal underlying data set as either Supporting Information files or to a stable, public repository and include the relevant URLs, DOIs, or accession numbers within your revised cover letter. For a list of acceptable repositories, please see http://journals.plos.org/plosone/s/data-availability#loc-recommended-repositories. Any potentially identifying patient information must be fully anonymized. Important: If there are ethical or legal restrictions to sharing your data publicly, please explain these restrictions in detail. Please see our guidelines for more information on what we consider unacceptable restrictions to publicly sharing data: http://journals.plos.org/plosone/s/data-availability#loc-unacceptable-data-access-restrictions . Note that it is not acceptable for the authors to be the sole named individuals responsible for ensuring data access. We will update your Data Availability statement to reflect the information you provide in your cover letter.

Answer: We have explained the ethical restrictions of sharing data that contains facial skin publicly and included this information in the cover letter:” This study data cannot be shared publicly because the scans contains the patient facial skin that could allow facial recognition. Data are available from the University of Michigan Ethics Committee (contact via luciacev@umich.edu) for researchers who meet the criteria for access to confidential data.”

Answer: We have now clarified that: ”All patient HIPAA identifiable personal information was removed from the DICOM files metadata through an anonymization process in the 3D Slicer platform . The anonymization was performed before the clinical centers shared the data for this retrospective study.The University of Michigan Institutional Review Board HUM00217585 waived the requirement for informed consent and granted IRB exemption. The patients' skin of the large field of scans was not removed; however those files are used only for training of the proposed machine learning model.”

For figure 1, the patient has written consent on file for the use of the images acquired for clinical purposes.

Reviewer #1: I have read the manuscript “Automatic multi-anatomical skull structure segmentation of

cone-beam computed tomography scans using 3D UNETR”. I have following comments.

1. The literature survey is poor and the relevant algorithms are required to be cited in the paper.

Answer: We have now added a discussion section and cited 17 additional references to relevant algorithms.

2. The introduction of the proposed coarse-to-fine framework is simple, especially about the patch-based semantic segmentation. As the main method of this paper, it would be better to give a more detail explanation about the coarse-to-fine framework and patch-based semantic segmentation.

Answer: We have clarified the introduction and we have now provided more detail of the proposed algorithm framework (Modification of the method section and a new UNETR section)

3. The results from the relevant algorithms (of airway segmentation) are required to be compared with the proposed method results.

Answer: We have now added to the discussion section comparison with studies on segmentation algorithms for various anatomic structures:” Recent studies have focused on only one specific facial structure such as the maxilla, [17], mandible [18] or airway [12], and used smaller samples from a single CBCT acquisition protocol, thus, those algorithms are not yet generalizable like the proposed AMASSS.”

4. U-Net Architecture was used in the manuscript. However, authors didn't disclose the details of the deep learning architecture.

Answer: We have now added a subsection UNETR in the proposed method.

5. No result was shown for the comparison of manual and automatic segmentation which was claimed through the title. The results are required which can show the error/difference between the manual volume assessment and automatic volume assessment.

Answer: Table 2 results present the comparison of manual and automatic segmentation. We added some descriptions of the formula used to compare. 

6. Again Literature is incomplete. There are many more articles on CBCT based volumetric segmentation and written much elaborately. Should have had a proper comparative table on literature and flow chart giving the flow of data analysis.

Answer: We have now improved the related works section, added a discussion section to compare 17 additional references to relevant algorithms.

7. Results are not discussed with any available literature. There lacks of conclusions and analysis of the experimental results in the results section. It would be more convincible if the authors can give additional technical analysis about their experimental results.

Answer: Response above. Thank you for your constructive comments. 

Reviewer #2: 

1.Introduction section needs to be re-written to improve its quality and readability.

Answer: Thank you for your helpful suggestion. We have re-written the introduction to improve its readability. And clarify important points to describe our objectives.

2.What is the motivation of the proposed work? Research gaps, objectives of the proposed work should be clearly justified

Answer: The introduction now highlights the research gaps and specific objectives of this study. 

3.Overall, the basic background is not introduced well, where the notations are not illustrated much clear.

Answer: We have re-written the introduction to improve its readability.

4.The literature has to be strongly updated with some relevant and recent papers focused on the fields dealt with in the manuscript.

Answer: We have now improved the related works section, added a discussion section to compare 17 additional references to more recent and relevant algorithms.

5.The study lacks a theoretical framework which is important for the reader to grasp the crust of the research.

Answer: We are confident that the revised introduction, proposed method and discussion sections address this comment. A new subsection UNETR has been added as well as a detail of the formula used in the training and result section.

6. Explain why the current method was selected for the study, its importance and compare with traditional methods.

Answer: The introduction section now better explain why the current method was selected for the study, its importance as well as a comparison with other methods in the related work and discussion section.

7.Authors are suggested to include more discussion on the results and also include some explanation regarding the justification to support why the proposed method is better in comparison towards other methods

Answer: We have now improved the related works section, added a discussion section to compare 17 additional references to more recent and relevant algorithms.

8.Does this kind of study have never attempted before? Justify this statement and give an appropriate explanation to do so in this paper.

Answer: We have now added a discussion section on the original contribution of this study. 

9.Quality of figures is so important too. Please provide some high-resolution figures. Some figures have a poor resolution.

Answer: We have submitted high-resolution figure in .tiff format that can be downloaded for full resolution visualization.

10.The language usage throughout this paper need to be improved, the author should do some proofreading on it.

Answer: Thank you. The article has now been proof-read by a native English speaker co-author.

---

## [Decision Letter · Decision Letter 1]

9 Sep 2022

Automatic multi-anatomical skull structure segmentation of cone-beam computed tomography scans using 3D UNETR

PONE-D-22-13724R1

Dear Dr. Gillot,

We’re pleased to inform you that your manuscript has been judged scientifically suitable for publication and will be formally accepted for publication once it meets all outstanding technical requirements.

Kind regards,

Sathishkumar V E

Academic Editor

PLOS ONE

Additional Editor Comments (optional):

Reviewers' comments:

Reviewer's Responses to Questions

**Comments to the Author**

1. If the authors have adequately addressed your comments raised in a previous round of review and you feel that this manuscript is now acceptable for publication, you may indicate that here to bypass the “Comments to the Author” section, enter your conflict of interest statement in the “Confidential to Editor” section, and submit your "Accept" recommendation.

Reviewer #1: All comments have been addressed

Reviewer #2: All comments have been addressed

2. Is the manuscript technically sound, and do the data support the conclusions?

Reviewer #1: Yes

Reviewer #2: Partly

3. Has the statistical analysis been performed appropriately and rigorously? 

Reviewer #1: Yes

Reviewer #2: N/A

4. Have the authors made all data underlying the findings in their manuscript fully available?

Reviewer #1: Yes

Reviewer #2: No

5. Is the manuscript presented in an intelligible fashion and written in standard English?

Reviewer #1: Yes

Reviewer #2: Yes

6. Review Comments to the Author

Reviewer #1: The relevant changes were made. Thus the paper is improved. The paper can be accepted as an original contribution.

Reviewer #2: 1.Identified research gaps and contribution of the proposed study should be elaborated.

2.The Limitations of the proposed study need to be discussed before conclusion.

3.What assumptions authors made during the simulation phase of this research work? Provide a critique on this aspect.

7. PLOS authors have the option to publish the peer review history of their article (what does this mean?). If published, this will include your full peer review and any attached files.

Reviewer #1: No

Reviewer #2: No

---

## [Editor Report · Acceptance letter]

21 Sep 2022

PONE-D-22-13724R1 

Automatic multi-anatomical skull structure segmentation of cone-beam computed tomography scans using 3D UNETR 

Dear Dr. Gillot:

I'm pleased to inform you that your manuscript has been deemed suitable for publication in PLOS ONE. Congratulations! Your manuscript is now with our production department. 

Kind regards, 

on behalf of

Dr. Sathishkumar V E 

Academic Editor

PLOS ONE